# Toward Escaping Model Collapse: Aligning Generated Images as a New Modality

## Abstract

Generative models have made it possible to synthesize highly realistic images, potentially providing an abundant data source for training machine learning models. Despite the advantages of these synthesizable data sources, the indiscriminate use of generated images as real images for training can harm model performance and even cause model collapse due to modality discrepancies between real and synthetic domains. In this paper, we propose a novel framework for discriminative use of generated images, coined *GenRA* (**Gen**erated-**R**eal **A**lignment), that explicitly treats generated images as a separate modality from real images. Instead of indiscriminately replacing real images with generated ones in the input space, our approach bridges the two distinct modalities in the same latent space through a multi-modal learning approach. To be specific, we first fine-tune a model exclusively on generated images using a cross-modality alignment loss and then employ this aligned model to further train various vision-language models with generated images. By aligning the two modalities, our approach effectively leverages the benefits of recent advances in generative models, thereby boosting the effectiveness of generated image training across a range of vision-language tasks. Our framework can be easily incorporated with various vision-language models, and we demonstrate its efficacy throughout extensive experiments. For example, our framework significantly improves performance on image captioning, zero-shot image retrieval, zero-shot image classification, and long caption retrieval tasks. It also shows positive generated data scaling trends and notable enhancements in the captioning performance of the large multimodal model, LLaVA.

## 1 Introduction

Generative models, such as GANs (Goodfellow et al., 2014; Chen et al., 2016) and diffusion models (Song et al., 2021a; Dhariwal & Nichol, 2021; Rombach et al., 2022), have revolutionized the field of computer vision by enabling the synthesis of highly realistic images. These generated images offer a rich and scalable source of data, which can significantly augment training datasets, enhance data diversity, and reduce the dependency on costly real-world data collection. However, despite their potential, incorporating generated images directly into training pipelines poses substantial challenges due to inherent modality discrepancies between generated and real images. This misalignment often leads to a phenomenon known as model collapse (Shumailov et al., 2024), where the model's performance severely deteriorates due to an over-reliance on generated content that fails to generalize well to real-world scenarios. To prevent model collapse in recursive scenarios, it is essential to first solve the gen-real discrepancy problem.

Existing approaches (Tian et al., 2023) typically integrate generated images into the training process without adequately addressing the modality gap between generated and real images. The resulting models are prone to overfitting the peculiarities of synthetic data, which negatively impacts performance across various downstream tasks, particularly when the model encounters real-world data. The primary source of this collapse lies in the failure to recognize that generated images, despite their realism, represent a distinct data modality that deviates from real images in subtle but significant ways. Addressing this modality gap is crucial to harnessing the full potential of generated data while maintaining robust performance on real-world tasks.

The challenge of using generated images stems from the fundamental differences between generated and real-world data distributions. Even when generated images appear visually convincing, they often contain subtle artifacts, biases, or domain-specific noise introduced during the generation process. These discrepancies are not just visual but can also affect higher-level semantic representations, resulting in a misalignment in the feature space that can propagate through the training pipeline. Furthermore, generative models may inadvertently capture and amplify biases present in their training data, leading to synthetic images that deviate in unexpected ways from real-world distributions. This modality gap poses significant challenges for downstream tasks, where models trained on misaligned data struggle with overfitting to generated features, reduced robustness, and degraded performance when applied to real images. Bridging this gap is critical to leveraging the strengths of generative models while avoiding pitfalls that compromise model reliability.

To tackle this challenge, we introduce a novel framework for **Gen**erated-**R**eal **A**lignment, namely *GenRA*, that explicitly treats generated images as a separate modality from real images. Unlike conventional methods that mix generated and real data indiscriminately, our approach bridges the two distinct modalities in the latent space by embedding generated images alongside real images having the same descriptions. Specifically, we fine-tune a model exclusively on generated images using a cross-modality alignment loss while keeping the pre-trained model for real images unchanged. This allows for explicit and adaptive alignment between the two modalities, enabling us to utilize the aligned model for training various vision-language models (Radford et al., 2021; Liu et al., 2023; Zhang et al., 2024) with highly realistic generated images. Thereby, we fully exploit the advantages of recent advances in generative models (Rombach et al., 2022), enhancing the performance of generated image training across various vision-language tasks.

Through the extensive experiments across a wide range of vision-language tasks, we demonstrate tje effectiveness of our framework by incorporating it with various vision-language models such as LLaVA (Liu et al., 2023). For example, our approach enhances image captioning on COCO (Lin et al., 2014), zero-shot image retrieval on COCO (Lin et al., 2014) and Flickr30k (Young et al., 2014), zero-shot image classification across eight widely used datasets, and long caption retrieval on ShareGPT4V (Chen et al., 2023). Furthermore, we observe positive generated data scaling trends in our framework across diverse datasets such as COCO (Lin et al., 2014), CC3M (Sharma et al., 2018), and CC12M (Changpinyo et al., 2021), highlighting the scalability of our method. Notably, our approach also improves the captioning performance of the recent large multimodal model, LLaVA (Liu et al., 2023), demonstrating its broad compatibility.

Our main contributions are summarized as:

- We introduce a novel framework for discriminative use of generated images, explicitly treating them as a distinct modality and aligning them with real images within the same latent space. It enables researchers to exploit highly realistic generated images effectively.

- We demonstrate the effectiveness of our framework through extensive experiments on a diverse set of vision-language benchmarks, including image captioning, zero-shot image retrieval, and zero-shot image classification, and further validate its compatibility with the recent large multimodal model, LLaVA.

- We explore the generated data scaling trend of our framework using large-scale generated datasets, demonstrating that our approach consistently improves as the volume of training data increases.

## 2 RELATED WORK

**Diffusion Models.** Diffusion models (Ho et al., 2020; Song et al., 2021b;a) have emerged as a powerful class of generative models, capable of producing high-quality images that closely mimic the distribution of real-world images. Prominent examples include Stable-Diffusion (Rombach et al., 2022), DreamBooth (Ruiz et al., 2022; 2023), and the DALL-E series (Ramesh et al., 2021; 2022; Betker et al., 2023), which have demonstrated remarkable success in generating diverse and complex images from textual descriptions. These models leverage advanced diffusion processes to iteratively refine images from noise, capturing intricate details and generating visually convincing outputs that can closely resemble real-world imagery. Our work utilizes the power of diffusion models to generate images, offering an innovative and cost-effective source of training data derived from textual

descriptions. By aligning these generated images with real image modalities through our GenRA framework, we bridge the gap between synthetic image generation and practical machine learning applications, addressing the challenges of model collapse due to modality discrepancies. This application of diffusion models represents a novel contribution to the field, as it not only enhances training efficiency but also expands the use of generative models beyond mere content creation, embedding them directly into the model training process to improve real-world performance.

**Generated Visual Learning.** Generated visual learning has gained traction as researchers explore the potential of synthetic data to augment traditional training paradigms. SynCLR (Tian et al., 2023) proposed a self-supervised framework that employs synthetic data to pre-train visual representations, demonstrating that models trained on generated data can achieve competitive results compared to those trained on real data. However, a critical challenge in this domain is the issue of model collapse, where the over-reliance on synthetic data without proper alignment leads to performance degradation when models are applied to real-world tasks. Recent work (Shumailov et al., 2024) highlights the inherent risks of training models on recursively generated data, emphasizing that models can inherit and amplify errors present in synthetic data, ultimately compromising their ability to generalize. Our research directly addresses these challenges by proposing a novel strategy that treats generated images as a distinct modality and aligns them with real images in the same latent space. This approach not only mitigates the risk of collapse but also enhances the robustness of models by embedding generated images within the same latent space as real images.

**Vision-Language Models.** Vision-language models, such as CLIP (Radford et al., 2021), have revolutionized cross-modal understanding by learning joint representations of images and text through contrastive learning objectives. While these models excel at leveraging large-scale real-world data, they often struggle when trained on generated images due to the modality gap. To overcome this, recent methods have explored various alignment techniques to improve cross-modal performance. For example, Long-CLIP (Zhang et al., 2024) extended CLIP by integrating longer captions, improving its ability to handle more descriptive textual inputs. Similarly, LLaVA (Liu et al., 2023) has demonstrated the potential for vision-language models to handle multimodal tasks like visual question answering and captioning by leveraging large-scale vision-language data. Our work builds on these foundational efforts by introducing an explicit generated-real alignment framework that enhances the adaptability of vision-language models when using generated data. By embedding generated images within the same latent space as real images and training the alignment, our approach directly addresses the modality discrepancies that limit model performance, offering a scalable solution that significantly boosts cross-modal learning across diverse vision-language tasks, including image captioning, zero-shot retrieval, and classification.

## 3 METHOD

In this section, we describe our proposed **Gen**erated-**R**eal **A**lignment (*GenRA*) framework, which tackles the challenge of training on generated images while ensuring robust performance during inference on real-world data, as illustrated in Figure 1. Our approach introduces two key components: (1) a Gen-CLIP flow on training and inference that handles generated and real images as separate modalities, and (2) an explicit alignment strategy with vision-language models to facilitate better integration with large language models (LLMs) such as CLIPCap (Mokady et al., 2021), LLaVA (Liu et al., 2023), and Llama3 (Meta, 2024). In this part, we detail the problem setup, the key components of our framework, and the alignment strategy used to enhance the performance of models trained on both generated and real data.

### 3.1 PRELIMINARIES

In this subsection, we introduce the problem setup and notations, followed by an overview of the contrastive language-image pre-training methodology that forms the foundation of our approach.

**Problem Setup and Notations.** Let $\mathcal{D}_r = \{(x_r, y_r)\}$ represent a dataset of real images with corresponding labels or annotations, and $\mathcal{D}_g = \{(x_g, y_g)\}$ denote a dataset of generated images synthesized by generative models, such as GANs or diffusion models. Our objective is to train a model $f(\cdot)$ that performs well across a broad set of downstream tasks, utilizing both $\mathcal{D}_r$ and $\mathcal{D}_g$, while mitigating the risk of model collapse caused by the inherent modality gap between $\mathcal{D}_r$ and $\mathcal{D}_g$. To formally

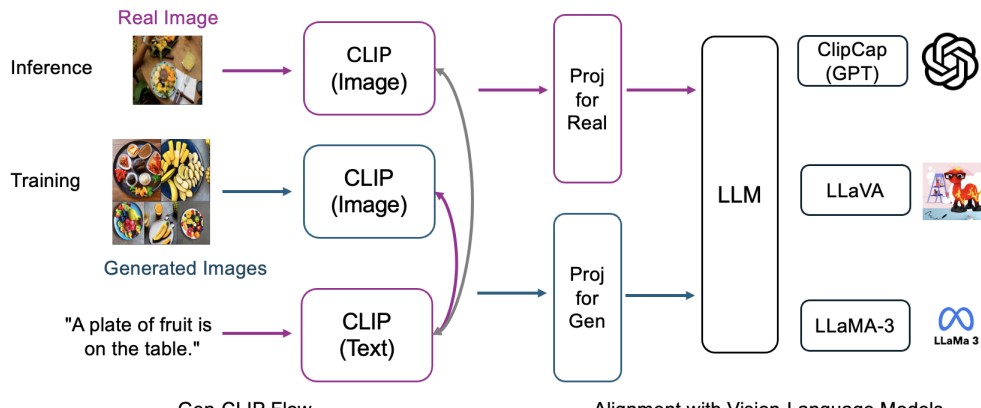

Figure 1: **Illustration of the proposed GenRA framework for vision-language tuning with gen-real alignment from diffusion models.** We introduce explicit alignment into the training regimen of the pre-trained CLIP model from real images to align the generated images with the real captions for training with state-of-the-art vision-language models.

define the alignment process, we introduce two models: a base model $f_r$, pre-trained on real images, and a fine-tuned model $f_g$, trained specifically on generated images. The primary goal of our framework is to align $f_g$ with $f_r$, ensuring that the feature representations of generated images are semantically consistent with those of real images. This alignment facilitates a unified understanding of both modalities, allowing the model to generalize across real data during inference.

**Contrastive Language-Image Pre-training.** Our framework builds on the foundation of Contrastive Language-Image Pre-Training (CLIP) (Radford et al., 2021), which learns joint embeddings for images and textual descriptions. CLIP leverages a contrastive loss that brings the embeddings of paired images and texts closer, while pushing apart the embeddings of unpaired ones, fostering cross-modal alignment. However, traditional CLIP training does not explicitly address the discrepancy between generated and real images, often leading to performance degradation when integrating generated data directly. To extend CLIP to handle generated images as a distinct modality, we propose a modified training objective that incorporates contrastive learning not only between real images and text but also between generated images and text. This treats generated and real images independently, preserving the unique characteristics of each modality during training.

### 3.2 GEN-CLIP FLOW

The first key component of our method is the *Gen-CLIP* flow, which focuses on training the model on generated images while treating them as a distinct modality. Unlike traditional approaches that mix generated and real images indiscriminately, we handle generated images separately to prevent the model from overfitting to the peculiarities of synthetic data. In the *Gen-CLIP* flow, we fine-tune a pre-trained CLIP model (Radford et al., 2021) using generated images, paired with the same textual descriptions used for real images. During fine-tuning, we employ a cross-modality alignment loss to minimize the feature space discrepancy between generated and real images. This contrastive alignment loss encourages the model to learn representations that place generated and real images with the same descriptions close to each other in the latent space, while maintaining their distinct modality-specific characteristics. To maintain computational efficiency and prevent catastrophic forgetting of real image representations, we apply Low-Rank Adaptation (LoRA) (Hu et al., 2021) during fine-tuning. LoRA introduces lightweight, efficient updates to the model, ensuring that the alignment process does not degrade the model's ability to generalize across different data modalities.

In the inference phase, the model fine-tuned on generated images in the *Gen-CLIP* flow is deployed to process real images without further fine-tuning. By keeping the pre-trained CLIP model for real images unchanged during the generated image training process, we ensure that the learned representations from the generated data remain aligned with real data. The *CLIP* flow leverages these aligned representations for inference on real images, allowing the model to generalize well to real-world data without suffering from the typical model collapse associated with over-reliance on generated content. This dual-model structure allows the model to benefit from the complementary strengths

of both real and generated images, ensuring that it performs robustly during real-world deployment while still benefiting from the scalability of generated training data. Note that the encoder fine-tuned with the LoRA and the projection for real is used on real images during inference time.

## 3.3 ALIGNMENT WITH VISION-LANGUAGE MODELS

Our alignment strategy is designed to enhance the integration of generated data into vision-language models, particularly large language models (LLMs) such as CLIPCap (Mokady et al., 2021), LLaVA (Liu et al., 2023), and Llama3 (Meta, 2024). This extension of GenRA ensures that generated images can be utilized effectively within these models for tasks such as image captioning, retrieval, and long-form question answering.

**Gen-Real Alignment.** The key to our framework is the cross-modality alignment loss, which ensures that generated images are embedded within the same latent space as real images, while maintaining their distinct characteristics. The alignment loss is formulated as:

$$\mathcal{L}_{align} = -\frac{1}{|\mathcal{B}|} \sum_{(x_g, x_r) \in \mathcal{B}} \log \frac{\exp(\text{sim}(f_g(x_g), f_r(x_r))/\tau)}{\sum_{x_r' \in \mathcal{B}} \exp(\text{sim}(f_g(x_g), f_r(x_r'))/\tau)}, \tag{1}$$

where $x_g$ and $x_r$ represent generated and real images, $f_g$ and $f_r$ are their corresponding feature representations, $\text{sim}(\cdot, \cdot)$ denotes cosine similarity between embeddings, and $\tau$ is a temperature parameter. This loss encourages generated images to be aligned with their real counterparts, facilitating effective transfer of knowledge across both modalities.

**CLIPCap (Mokady et al., 2021)** combines CLIP's image embeddings with a transformer-based language model to generate captions from images. By aligning generated images with real image embeddings, we ensure that CLIPCap can generate high-quality captions from both real and generated data. Fine-tuning CLIPCap with our alignment framework allows the model to handle both modalities effectively, resulting in enhanced performance on image captioning tasks.

**LLaVA (Liu et al., 2023) & Llama3 (Meta, 2024)** are advanced multimodal models designed to perform vision-language tasks. To align generated images with these models, we first fine-tune the vision-language models using our GenRA strategy to ensure that representations from generated data are aligned with real data. The aligned vision representations are then integrated with the LLMs, allowing the models to handle complex vision-language tasks such as long captioning and retrieval more effectively. This alignment enhances the robustness and flexibility of LLaVA and Llama3 in real-world applications involving both real and generated images.

Our framework is designed to scale effectively with larger datasets, as evidenced by the performance improvements observed on large-scale datasets such as CC3M (Sharma et al., 2018) and CC12M (Changpinyo et al., 2021). The alignment strategy ensures that as the volume of generated training data increases, the model continues to generalize effectively to real-world data. This scalability demonstrates the potential of GenRA as a cost-effective solution for training robust vision-language models using synthetic data.

## 4 EXPERIMENTS

In this section, we provide the experimental setup, evaluation metrics, and comparative analysis conducted to validate the effectiveness of our proposed method. Through rigorous experimentation on a diverse set of datasets, we assess our model's performance on image captioning, zero-shot image retrieval, and zero-shot image classification tasks, comparing it against existing benchmarks to highlight our contributions.

### 4.1 EXPERIMENTAL SETUP

**Datasets.** Our experiments leverage a comprehensive collection of datasets to evaluate the versatility and effectiveness of our proposed Gen-Real alignment framework. We focus on a diverse set of tasks, including image captioning, zero-shot image retrieval, and zero-shot image classification, ensuring broad coverage across various domains. **COCO** (Lin et al., 2014): We use the COCO dataset for image captioning and zero-shot image retrieval tasks, as it provides a diverse collection

Table 1: **Image captioning.** We perform semi-images fine-tuning on pre-trained ClipCap and LLaMA-3 for image captioning on COCO. We report the standard metrics to evaluate the quality of generated captions. The best results are indicated in **bold**.

| Method | B@4 (↑) | METEOR(↑) | CIDEr (↑) | SPICE (↑) | ROUGE-L (↑) | WMD (↑) |
|---|---|---|---|---|---|---|
| ClipCap (Mokady et al., 2021) | 32.15 | 27.10 | 108.35 | 20.12 | – | – |
| ClipCap + GenRA (ours) | **38.12** | **31.67** | **119.53** | **23.75** | **56.27** | **62.16** |
| LLaVA (Liu et al., 2023) | 39.67 | 32.38 | 134.29 | 24.17 | 61.36 | 65.78 |
| LLaVA + GenRA (ours) | **43.26** | **34.89** | **146.38** | **27.23** | **65.25** | **71.39** |
| Llama3 (Meta, 2024) | 47.36 | 35.21 | 158.13 | 28.35 | 68.32 | 75.13 |
| Llama3 + GenRA (ours) | **50.21** | **38.59** | **168.53** | **32.58** | **73.29** | **80.25** |

of real-world images paired with detailed captions, which serve as a benchmark for evaluating the alignment between generated and real image modalities. **Zero-Shot Image Classification**: Following the original CLIP (Radford et al., 2021) setup, we evaluate our model on eight widely-used benchmarks to assess its performance across diverse visual recognition tasks: **DTD** (Cimpoi et al., 2014): evaluates the model's ability to classify textural attributes in images. **Stanford Cars** (Krause et al., 2013): a fine-grained visual classification task focusing on car models. **SUN397** (Xiao et al., 2010; 2014): a large-scale scene classification dataset that tests the model's scene understanding capabilities. **Food 101** (Bossard et al., 2014): assesses the model's ability to recognize food items from various cuisines. **Aircraft** (Maji et al., 2013): a dataset for fine-grained classification of aircraft models. **Oxford Pets** (Parkhi et al., 2012): used for breed classification of cats and dogs. **Caltech 101** (Fei-Fei et al., 2004): a general object recognition dataset covering a wide range of categories. **ImageNet 1K** (Deng et al., 2009): a large-scale benchmark for object classification tasks. **CC3M** (Sharma et al., 2018) and **CC12M** (Changpinyo et al., 2021): To demonstrate the scaling behavior of our Gen-Real alignment approach, we include large-scale datasets CC3M and CC12M, allowing us to explore the effectiveness of our method when training with extensive generated and real image collections. **ShareGPT4V**: For long caption retrieval, we utilize ShareGPT4V, which challenges the model to handle complex, descriptive captions associated with generated and real images, emphasizing the need for strong cross-modal alignment.

**Evaluation Metrics.** To comprehensively evaluate our framework, we employ task-specific metrics tailored to image captioning, zero-shot image retrieval, and zero-shot image classification: **Image Captioning**: Performance is assessed using standard metrics such as BLEU@4 (B@4) (Papineni et al., 2002), METEOR (Denkowski & Lavie, 2014), CIDEr (Vedantam et al., 2014), SPICE (Anderson et al., 2016), ROUGE-L (Lin & Och, 2004), and Word Mover's Distance (WMD) (Kusner et al., 2015). These metrics evaluate the quality and semantic accuracy of generated captions compared to ground truth. **Zero-Shot Image Retrieval**: We measure both image-to-text and text-to-image retrieval capabilities using Recall@1, Recall@5, and Recall@10. These metrics assess the model's ability to correctly retrieve relevant items based on the provided query, highlighting its cross-modal understanding. **Zero-Shot Image Classification**: The classification performance on unseen categories is evaluated using top-1 accuracy, reflecting the model's generalization ability to new classes without prior training on those specific categories.

**Implementation.** For image captioning, we adhere to the implementation strategy of ClipCap (Mokady et al., 2021), which combines CLIP with a text generation model to produce descriptive captions for images. ClipCap uses CLIP's image embeddings as input to a transformer-based captioning model, enabling the generation of semantically accurate and contextually rich captions for both real and generated images. For zero-shot evaluation on both retrieval and image classification tasks, we follow the setup detailed in the original CLIP (Radford et al., 2021) paper. This setup emphasizes the model's ability to generalize across unseen data by using natural language prompts to guide image classification and retrieval, leveraging the contrastive training between images and textual descriptions without explicit fine-tuning on target datasets. We adopt Stable Diffusion v2 (Rombach et al., 2022) to generate semi-images using captions from the COCO (Lin et al., 2014) train2014 set. Stable Diffusion provides high-quality image synthesis, enabling us to produce generated images that are both visually realistic and semantically aligned with the training captions, serving as the generated modality in our alignment framework. During fine-tuning, we use a rank of 4 in Low-Rank Adaptation (LoRA) to adjust the model parameters specifically for generated images, ensuring that the adaptation remains efficient and computationally manageable. LoRA fine-tuning allows us to modify the model with a minimal increase in computational overhead, preserving the

Table 2: **Zero-shot image retrieval on COCO.** We perform zero-shot retrieval on pre-trained Semi-CLIP for image retrieval on the COCO benchmark. We report the image-to-text and text-to-image Recall@1,5,10 metrics to evaluate the quality of retrieved images.

| Method | Image-to-Text | | | Text-to-Image | | |
|---|---|---|---|---|---|---|
| | R@1 (↑) | R@5 (↑) | R@10 (↑) | R@1 (↑) | R@5 (↑) | R@10 (↑) |
| CLIP (Radford et al., 2021) | 51.8 | 76.8 | 84.3 | 32.7 | 57.7 | 68.2 |
| CLIP + GenRA (ours) | **56.8** | **80.1** | **87.2** | **37.5** | **62.7** | **73.2** |
| Long-CLIP (Zhang et al., 2024) | 57.2 | 80.8 | 87.8 | 40.4 | 65.9 | 75.7 |
| Long-CLIP + GenRA (ours) | **62.3** | **84.1** | **91.2** | **45.6** | **69.8** | **79.5** |

Table 3: **Zero-shot image retrieval on Flickr30k.** We perform zero-shot retrieval on pre-trained SemiCLIP for image retrieval on the Flickr30k benchmark. We report the image-to-text and text-to-image Recall@1,5,10 metrics to evaluate the quality of retrieved images.

| Method | Image-to-Text | | | Text-to-Image | | |
|---|---|---|---|---|---|---|
| | R@1 (↑) | R@5 (↑) | R@10 (↑) | R@1 (↑) | R@5 (↑) | R@10 (↑) |
| CLIP (Radford et al., 2021) | 44.1 | 68.2 | 77.0 | 24.7 | 45.1 | 54.6 |
| CLIP + GenRA (ours) | **47.1** | **71.2** | **79.6** | **30.2** | **50.3** | **60.5** |
| Long-CLIP (Zhang et al., 2024) | 47.2 | 71.5 | 80.0 | 33.1 | 55.6 | 64.9 |
| Long-CLIP + GenRA (ours) | **51.6** | **75.3** | **83.6** | **39.3** | **61.5** | **71.8** |

model's core capabilities while enhancing its alignment with the generated data. For optimization, we use the AdamW optimizer with a learning rate of $1 \times 10^{-4}$ and weight decay of 0.01. We employ a cosine annealing schedule with warm restarts to dynamically adjust the learning rate, enhancing convergence stability across training phases. Batch normalization and gradient clipping are applied to prevent exploding gradients and ensure smooth training dynamics.

## 4.2 COMPARISON TO PRIOR WORK

**Image Captioning.** We compare our model's performance on the COCO dataset against prior commonly-used baselines, including ClipCap (Mokady et al., 2021), LLaVA (Liu et al., 2023), and LLAMA-3 (Meta, 2024) The results, detailed in Table 1, demonstrate significant improvements across all evaluated metrics, underscoring the efficacy of our Gen-Real Alignment (GenRA) approach when combined with semi-images and LoRA optimization. For ClipCap, the proposed ClipCap + GenRA configuration achieves 38.12 B@4, 31.67 METEOR, 119.53 CIDEr, 23.75 SPICE, 56.27 ROUGE-L, and 62.16 WMD, significantly outperforming the baseline ClipCap and the ClipCap + LoRA setup. Specifically, our GenRA approach boosts the original ClipCap (Mokady et al., 2021) by 5.97 B@4, 4.57 METEOR, 11.18 CIDEr, and 3.63 SPICE. These results highlight the advantages of aligning generated and real images within a unified semantic space, allowing for enhanced image captioning performance. Similarly, when applied to LLAMA-3, our LLAMA-3 + GenRA model reaches 50.21 B@4, 38.59 METEOR, 168.53 CIDEr, 32.58 SPICE, 73.29 ROUGE-L, and 80.25 WMD, demonstrating notable improvements over both the baseline and the LoRA fine-tuning strategy. Compared to LLAMA-3 alone, GenRA achieves gains of 2.85 B@4, 2.46 METEOR, 10.35 CIDEr, and 4.30 SPICE, establishing our approach as a robust technique for enhancing models through gen-real alignment. The substantial gains observed across both model architectures confirm the effectiveness of our GenRA framework. By fine-tuning with generated images while maintaining alignment with real image modalities, our method effectively bridges the modality gap, resulting in better understanding and generation of descriptive captions aligned with real-world data.

**Zero-shot Image Retrieval.** The comparative results in Tables 2 and 3 highlight our model's superior recall rates, showcasing its robustness in understanding and associating visual and textual data. Our method is evaluated on two benchmarks: COCO and Flickr30k, using both image-to-text and text-to-image retrieval tasks, demonstrating significant improvements over prior baselines. On the COCO dataset, our approach, CLIP + GenRA, achieves 56.8 R@1, 80.1 R@5, and 87.2 R@10 for image-to-text retrieval, outperforming the original CLIP (Radford et al., 2021) trained on real images by 5.0 R@1, 3.3 R@5, and 2.9 R@10. For text-to-image retrieval, CLIP + GenRA scores 37.5 R@1, 62.7 R@5, and 73.2 R@10, demonstrating gains of 4.8 R@1, 5.0 R@5, and 5.0 R@10 compared to the baseline CLIP. These improvements validate the effectiveness of our alignment strategy in bridging the gap between generated and real image modalities, enhancing zero-shot re-

Table 4: **Zero-shot image classification.** We perform a zero-shot evaluation on pre-trained Semi-CLIP for image classification on eight benchmarks. We report the top-1 accuracy to evaluate the quality of learned representations from semi-images. The best results are indicated in **bold**.

| Method | DTD | Stanford Cars | SUN397 | Food 101 | Aircraft | Oxford Pets | Caltech 101 | ImageNet |
|---|---|---|---|---|---|---|---|---|
| CLIP (Radford et al., 2021) | 55.20 | 77.53 | 69.31 | 93.08 | 32.88 | 93.33 | 93.24 | 75.54 |
| CLIP + GenRA (ours) | **65.26** | **81.32** | **75.53** | **95.21** | **37.85** | **95.23** | **95.57** | **77.68** |
| SynCLR (Tian et al., 2023) | 79.90 | 93.80 | 76.20 | 91.60 | 81.70 | 93.60 | 95.30 | 85.80 (ft) |
| SynCLR + GenRA (ours) | **83.67** | **96.56** | **81.25** | **96.38** | **86.75** | **95.70** | **98.35** | **87.95** (ft) |

Table 5: **Long caption retrieval on ShareGPT4V.** We report the image-to-text and text-to-image Recall@1 to evaluate the quality of retrieved images. The best results are indicated in **bold**.

| Method | Image-to-Text | Text-to-Image |
|---|---|---|
| CLIP (Radford et al., 2021) | 78.2 | 79.6 |
| CLIP + GenRA (ours) | **85.2** | **86.7** |
| Long-CLIP (Zhang et al., 2024) | 94.6 | 93.3 |
| Long-CLIP + GenRA (ours) | **97.2** | 96.1 |

trieval capabilities. Similarly, when applied to the Long-CLIP architecture (Zhang et al., 2024), our Long-CLIP + GenRA configuration further boosts performance, achieving 62.3 R@1, 84.1 R@5, and 91.2 R@10 on image-to-text retrieval, and 45.6 R@1, 69.8 R@5, and 79.5 R@10 on text-to-image retrieval. This demonstrates that GenRA consistently enhances model performance across different backbone architectures by facilitating better alignment of generated images with real-world data. On the Flickr30k dataset, our CLIP + GenRA model achieves 47.1 R@1, 71.2 R@5, and 79.6 R@10 for image-to-text retrieval, outperforming CLIP by 3.0 R@1, 3.2 R@5, and 2.6 R@10. In text-to-image retrieval, the model scores 39.3 R@1, 61.5 R@5, and 71.8 R@10, with respective gains of 14.6 R@1, 16.0 R@5, and 17.2 R@10 over CLIP. These results validate the robustness of our approach in learning meaningful representations from generated images for zero-shot retrieval on real images, highlighting the advantages of our Gen-Real alignment in enhancing cross-modal retrieval tasks across various benchmarks and model architectures.

**Zero-shot Image Classification.** We evaluate the zero-shot classification performance of our model across eight diverse benchmarks, including DTD, Stanford Cars, SUN397, Food 101, Aircraft, Oxford Pets, Caltech 101, and ImageNet 1K. As shown in Table 4, our model consistently achieves top-1 accuracy surpassing previous approaches, validating the advantage of leveraging generated images through our framework for enhancing zero-shot learning capabilities. Our CLIP + GenRA approach achieves a top-1 accuracy of 65.26 on the DTD benchmark, outperforming the original CLIP (Radford et al., 2021) by 10.06 points, demonstrating the significant benefit of aligning generated images with real data. On the Stanford Cars dataset, our model reaches 81.32, showing robust performance gains, particularly in fine-grained classification tasks. For the challenging FGVC Aircraft benchmark, our method scores 37.85, marking a substantial improvement of 4.97 over the baseline CLIP, highlighting our model's capacity to handle complex visual distinctions. Additionally, our model performs exceptionally well on other benchmarks, achieving 75.53 on SUN397, 95.21 on Food 101, 95.23 on Oxford Pets, 95.57 on Caltech 101, and 77.68 on ImageNet 1K. These results consistently outperform both the standard CLIP and the CLIP + LoRA setup, confirming the effectiveness of our gen-real alignment strategy in broadening the model's generalization capabilities across various domains. Through these experiments, we affirm the effectiveness of our methodology in advancing the state-of-the-art across a spectrum of visual and textual understanding tasks.

**Long Caption Retrieval.** We evaluate our model's capability to handle long captions using the ShareGPT4V (Chen et al., 2023) benchmark, as reported in Table 5. The evaluation focuses on image-to-text and text-to-image retrieval tasks, with Recall@1 used to assess the quality of retrieved results. Our model demonstrates an enhanced ability to comprehend and generate relevant responses to extended textual inputs, affirming its utility in applications that require detailed and descriptive outputs. For the CLIP-based models, our CLIP + GenRA configuration achieves 85.2 for image-to-text and 86.7 for text-to-image retrieval, outperforming both the original CLIP (Radford et al., 2021) and the CLIP + LoRA variants. This result highlights the effectiveness of our alignment strategy in bridging the semantic gap between generated and real images, particularly when handling complex, long-caption scenarios. When applied to the Long-CLIP architecture (Zhang et al., 2024),

Table 6: **Ablation study on Gen-Real Alignment.** We perform ablation studies on image captioning from pre-trained CLIP on generated images. The best results are indicated in **bold**.

| Alignment | B@4 (↑) | METEOR(↑) | CIDEr (↑) | SPICE (↑) | ROUGE-L (↑) | WMD (↑) |
|---|---|---|---|---|---|---|
| ✗ | 36.15 | 30.32 | 115.35 | 22.95 | 55.12 | 61.08 |
| ✓ | **38.12** | **31.67** | **119.53** | **23.75** | **56.27** | **62.16** |

Table 7: **Scaling trend of Gen-Real alignment on zero-shot image retrieval on Flickr30k.** We perform zero-shot retrieval on models trained from COCO, CC3M, and CC12M on the Flickr30k benchmark. We report the Recall@1,5,10 metrics to evaluate the quality of retrieved images.

| Train Data | Image-to-Text | | | Text-to-Image | | |
|---|---|---|---|---|---|---|
| | R@1 (↑) | R@5 (↑) | R@10 (↑) | R@1 (↑) | R@5 (↑) | R@10 (↑) |
| COCO | 47.1 | 71.2 | 79.6 | 30.2 | 50.3 | 60.5 |
| CC3M | 48.6 | 73.6 | 82.2 | 32.6 | 52.6 | 62.3 |
| CC12M | **50.9** | **75.3** | **84.6** | **34.9** | **54.7** | **64.8** |

our Long-CLIP + GenRA configuration reaches 97.2 for image-to-text and 96.1 for text-to-image retrieval, marking the highest performance among all tested configurations. These gains of 2.6 and 1.6 over Long-CLIP + LoRA confirm that our approach not only strengthens the alignment between modalities but also substantially improves the retrieval of images and captions involving extended and intricate descriptions. Overall, the results confirm the robustness and scalability of our framework in managing complex captioning tasks, paving the way for more nuanced and effective models in vision-language applications that involve detailed descriptive content.

## 4.3 EXPERIMENTAL ANALYSIS

In this section, we performed ablation studies to demonstrate the benefit of gen-real alignment. We also conducted extensive experiments to explore the scaling trend on different training data sizes.

**Gen-Real Alignment.** To quantify the impact of gen-real alignment fine-tuning on our model's performance, we conducted ablation studies comparing models with and without alignment optimization. The results, presented in Table 6, demonstrate significant improvements across all metrics when alignment tuning is applied, validating the effectiveness of our proposed approach. In the context of image captioning tasks, models fine-tuned with gen-real alignment consistently outperform their counterparts that lack this optimization step. Specifically, adding gen-real alignment to the vanilla baseline using semi-images to fine-tune all parameters led to substantial increases across all evaluated metrics: 3.56 in B@4, 1.13 in METEOR, 4.18 in CIDEr, 0.8 in SPICE, 1.15 in ROUGE-L, and 1.09 in WMD. These improvements highlight the critical role of alignment fine-tuning in bridging the modality gap between generated and real images, which enables the model to better capture and replicate the semantic richness found in real-world data. The results underscore the effectiveness of gen-real alignment in optimizing model performance, particularly in adapting to the nuances of semi-generated images and their associated textual descriptions. By embedding generated images within the same latent space as real images, our approach enhances the model's ability to understand and process complex visual-language relationships, ultimately leading to superior performance in downstream tasks.

**Scaling trend of Gen-Real alignment.** To further evaluate the scalability of our proposed Gen-Real alignment, we explore its performance across varying scales of training data. Specifically, we apply our training framework on semi-images derived from COCO Lin et al. (2014), CC3M Sharma et al. (2018), and CC12M Changpinyo et al. (2021). The comparison results on zero-shot image retrieval on the Flickr30k benchmark are reported in Table 7. The results reveal a clear scaling trend, where increasing the volume of training data from COCO to CC3M and then to CC12M consistently enhances the model's performance on both image-to-text and text-to-image retrieval tasks. Specifically, our model trained on CC12M achieves the highest scores with 50.9 R@1, 75.3 R@5, and 84.6 R@10 for image-to-text retrieval, and 34.9 R@1, 54.7 R@5, and 64.8 R@10 for text-to-image retrieval, outperforming the models trained on the smaller COCO and CC3M datasets. These improvements demonstrate that our Gen-Real alignment framework benefits significantly from larger and more diverse training datasets of generated images, effectively capturing richer semantic representations and enhancing retrieval capabilities. The results underscore the effectiveness of our

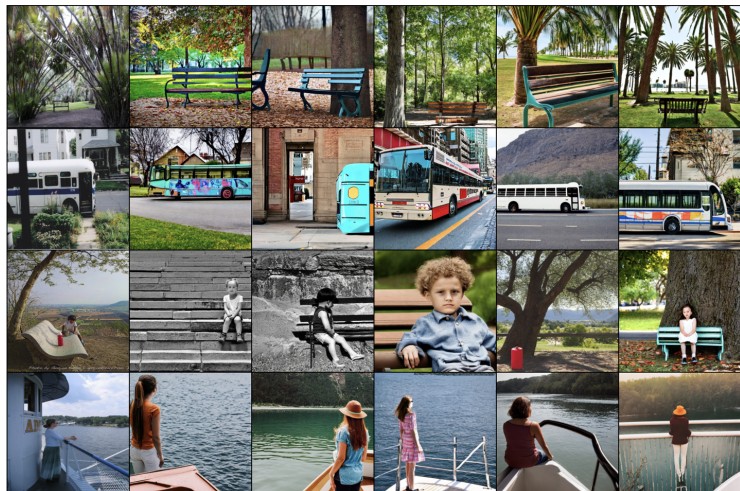

Figure 2: **Visualizations of real (Column 1) and generated images (Columns 2-6) using the same caption.** Those generated images generally capture high-level semantics in real images.

method in leveraging the scaling trend of generated data, showing that as the scale of semi-images increases, our model continues to learn and generalize better across zero-shot retrieval tasks.

**Visualization of Generated Images.** To further understand the quality and semantic alignment of the generated images used in our training process, we provide visualizations of a subset of generated images alongside their corresponding real-world counterparts, as shown in Figure 2. These images were generated using state-of-the-art generative models such as Stable Diffusion (Rombach et al., 2022), and are designed to closely match the real-world data in terms of visual realism and content. Through these visualizations, we observe that while generated images generally capture high-level features and structures present in real images, they may still exhibit subtle artifacts or variations that could contribute to the modality gap. Despite these differences, our Gen-Real Alignment framework successfully bridges this gap, as evidenced by the alignment of semantic features between the generated and real images in the learned latent space. The visualizations not only illustrate the potential of generated data as a cost-effective supplement to real-world data but also highlight the importance of explicit alignment strategies to mitigate discrepancies between generated and real data distributions.

## 5 CONCLUSION

In this work, we present GenRA, a novel framework for gen-real alignment that addresses the modality gap between generated and real images, a key challenge that often leads to model collapse when integrating generated data into training pipelines. Our approach explicitly treats generated images as a separate modality and employs a training scheme that aligns these images within the same latent space as real images. By fine-tuning models on generated images while maintaining a pre-trained model for real images, our framework facilitates explicit alignment between the two modalities, leading to significant performance improvements across various vision-language tasks. Extensive experiments demonstrate the efficacy of our method on a wide range of benchmarks, including image captioning, zero-shot image retrieval, and zero-shot image classification. Our results consistently show that GenRA enhances the model's ability to generalize and perform across tasks, particularly when trained on large-scale datasets. The scaling trend observed with larger generated datasets such as CC12M further highlights the robustness and adaptability of our approach.

**Limitation.** While our approach significantly improves the performance of models trained on generated images, it relies heavily on high-quality generative models that produce images with realistic and semantically accurate content.

**Broader Impact.** Our proposed Gen-Real alignment framework enhances the integration of generated images in machine learning, potentially reducing the dependency on costly and time-consuming real-world data collection. This has broad implications for democratizing access to high-quality training data, especially in fields where obtaining real data is challenging or ethically sensitive.

## ETHICS STATEMENT

Our work leverages generative models, such as stable diffusion models (Rombach et al., 2022), to create generated images that can supplement real-world datasets in training machine learning models. While this approach offers significant benefits in terms of reducing the need for expensive and time-consuming real-world data collection, we recognize the potential ethical risks associated with generated data. Generated images may inadvertently reflect biases present in the data used to train the generative models, potentially perpetuating harmful stereotypes or inaccuracies. To mitigate this, we emphasize the importance of careful curation of training datasets and encourage the community to develop strategies for auditing and debiasing generative models. Additionally, the alignment of generated data with real-world data must be handled with caution, as over-reliance on generated content can obscure important real-world variations.

## REPRODUCIBILITY STATEMENT

To ensure the reproducibility of our work, we have provided detailed descriptions of our experimental setup, datasets, and models in the Method and Experiments sections. Specifically, we describe the datasets used, including COCO, CC3M, CC12M, and Flickr30k, as well as the generative models (e.g., Stable Diffusion) employed to synthesize the semi-images. Additionally, we outline the key components of our framework, including the explicit alignment process, contrastive loss functions, and the model training strategy. For ease of reproducibility, we will release our code, model weights, and hyperparameters upon publication. We encourage the use of standardized benchmarks, as described in the paper, and provide detailed instructions on how to replicate the training and evaluation procedures for both generated and real images. Furthermore, we will ensure that all pre-trained models, including those fine-tuned on generated images, are accessible for evaluation by the broader research community. By making all resources publicly available, we aim to promote transparent and reproducible research in the integration of generated data with real-world training pipelines.

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

APPENDIX

In this appendix, we provide the following material:

- addition implementation and datasets details in Section A,
- algorithm for our GenRA in Section B,
- more discussions on Gen-Real alignment in Section C,
- more experimental analyses in Section D,
- qualitative visualization results in Section E,
- discussions on limitations and broader impact in Section F.

## A   IMPLEMENTATION & DATASET DETAILS

In this section, we provide additional implementation details to ensure the reproducibility of our experiments, along with a comprehensive description of the datasets used.

**Implementation.** The base model used in our framework is the CLIP model (Radford et al., 2021), pre-trained on real images and paired with their textual descriptions. We fine-tune the pre-trained CLIP model on generated images using the LoRA (Hu et al., 2021) method to introduce low-rank updates, ensuring that the training remains computationally efficient. For contrastive learning, we set the temperature parameter $\tau = 0.07$ and optimize using the AdamW optimizer with a learning rate of $1 \times 10^{-4}$ and a batch size of 256. The synthetic training data were generated using Stable Diffusion v2 on NVIDIA A100-80GB GPUs. The number of generated images is consistent with the number of text-image pairs in the original training set: 560k for COCO, 3.3 million for CC3M, and 12 million for CC12M. Each image was generated with 50 inference steps, balancing quality and computational efficiency. The total generation time is 5 GPU days for COCO, 30 GPU days for CC3M, and 109 GPU days for CC12M. Parallelized generation was employed for larger datasets like CC12M. Fine-tuning for "Proj for Real" and "Proj for Gen" was performed for 50,000 steps.

**Datasets.** To evaluate the versatility and effectiveness of our Gen-Real Alignment framework, we employ a comprehensive suite of datasets across a variety of tasks, including image captioning, zero-shot image retrieval, and zero-shot image classification. This ensures a broad assessment of our model's performance across multiple domains and challenges.

- **COCO** (Lin et al., 2014): The COCO dataset is used for both image captioning and zero-shot image retrieval tasks. It offers a large and diverse collection of real-world images paired with detailed textual descriptions, serving as a benchmark for evaluating the alignment of generated and real image modalities.

- **Zero-Shot Image Classification**: To evaluate the generalization capabilities of our model, we utilize eight well-known benchmarks, following the setup of the original CLIP (Radford et al., 2021):

  - **DTD** (Cimpoi et al., 2014): Tests the model's ability to classify textures across various images.
  - **Stanford Cars** (Krause et al., 2013): A dataset focusing on fine-grained classification of car models, used to assess the model's capacity to distinguish between visually similar objects.
  - **SUN397** (Xiao et al., 2010; 2014): A large-scale scene classification dataset used to evaluate scene understanding.
  - **Food 101** (Bossard et al., 2014): A benchmark used to assess the model's ability to classify food items from various cuisines.
  - **Aircraft** (Maji et al., 2013): Used for fine-grained classification of aircraft models, testing the model's accuracy in distinguishing similar objects.
  - **Oxford Pets** (Parkhi et al., 2012): A dataset focused on the classification of various pet breeds, including both dogs and cats.
  - **Caltech 101** (Fei-Fei et al., 2004): A widely used object recognition dataset covering a variety of general categories.

---

**Algorithm 1** GenRA Algorithm: Training and Inference on Generated and Real Images

---

**Require:** Datasets of real images $\mathcal{D}_r = \{(x_r, y_r)\}$ and generated images $\mathcal{D}_g = \{(x_g, y_g)\}$, pre-trained CLIP model $f_r$, learning rate $\eta$, batch size $|\mathcal{B}|$, temperature $\tau$, LoRA parameters.

**Ensure:** Fine-tuned model $f_g$ for generated images, aligned with $f_r$ for real images.

1: **Initialize:** Load the pre-trained CLIP model $f_r$ trained on real images, set the alignment loss as $\mathcal{L}_{align}$.

2: **Step 1: Gen-CLIP Flow for Training on Generated Images.**

3: **for** each mini-batch $\mathcal{B}_g$ from $\mathcal{D}_g$ **do**

4:     Extract image features $f_g(x_g)$ for each $x_g \in \mathcal{B}_g$ using the CLIP model $f_g$.

5:     Extract textual features $f_r(y_g)$ corresponding to each $x_g$ from the text encoder.

6:     Compute cross-modality alignment loss $\mathcal{L}_{align}$:

$$\mathcal{L}_{align} = -\frac{1}{|\mathcal{B}|} \sum_{(x_g, x_r) \in \mathcal{B}} \log \frac{\exp(\text{sim}(f_g(x_g), f_r(x_r))/\tau)}{\sum_{x'_r \in \mathcal{B}} \exp(\text{sim}(f_g(x_g), f_r(x'_r))/\tau)}$$

7:     Apply LoRA updates to minimize $\mathcal{L}_{align}$.

8:     Update model parameters $f_g \leftarrow f_g - \eta \nabla_{f_g} \mathcal{L}_{align}$.

9: **end for**

10: **Step 2: CLIP Flow for Inference on Real Images.**

11: **for** each mini-batch $\mathcal{B}_r$ from $\mathcal{D}_r$ **do**

12:     Extract real image features $f_r(x_r)$ using the pre-trained model $f_r$.

13:     Use the aligned representations from $f_g$ for inference on real images.

14: **end for**

15: **Step 3: Alignment with Vision-Language Models.**

16: **for** each LLM (e.g., CLIPCap, LLaVA, LLaMA3) **do**

17:     Fine-tune the LLM using the aligned generated and real image embeddings.

18: **end for**

19: **Return:** Aligned model $f_g$ for generated images, aligned with the real-image model $f_r$.

---

- **ImageNet 1K** (Deng et al., 2009): A benchmark for large-scale object classification, testing the model's ability to handle diverse image categories.

- **CC3M** (Sharma et al., 2018) and **CC12M** (Changpinyo et al., 2021): These large-scale datasets provide millions of image-caption pairs, allowing us to explore the scalability of our Gen-Real alignment framework. We evaluate our model's performance when trained on both real and generated data from these expansive datasets.

- **ShareGPT4V**: To evaluate long caption retrieval, we use the ShareGPT4V dataset, which includes complex and descriptive captions associated with both generated and real images. This dataset emphasizes the importance of strong cross-modal alignment for retrieving long, detailed captions.

**Evaluation Metrics.** To comprehensively evaluate our framework, we employ task-specific metrics tailored to image captioning, zero-shot image retrieval, and zero-shot image classification:

- **Image Captioning**: Performance is assessed using standard metrics such as BLEU@4 (B@4) (Papineni et al., 2002), METEOR (Denkowski & Lavie, 2014), CIDEr (Vedantam et al., 2014), SPICE (Anderson et al., 2016), ROUGE-L (Lin & Och, 2004), and Word Mover's Distance (WMD) (Kusner et al., 2015). These metrics evaluate the quality and semantic accuracy of generated captions compared to the ground truth.

- **Zero-Shot Image Retrieval**: We measure both image-to-text and text-to-image retrieval capabilities using Recall@1, Recall@5, and Recall@10. These metrics assess the model's ability to correctly retrieve relevant items based on the provided query, highlighting its cross-modal understanding.

- **Zero-Shot Image Classification**: Classification performance on unseen categories is evaluated using top-1 accuracy, which reflects the model's generalization ability to classify new classes without prior training on those specific categories.

This experimental setup allows us to thoroughly validate our Gen-Real alignment framework across a wide range of tasks, demonstrating its effectiveness in addressing the modality gap between generated and real images and enhancing performance across diverse vision-language applications.

## B  GENRA ALGORITHM

In this section, we outline the algorithm that implements the **Gen**erated-**R**eal **A**lignment (*GenRA*) framework, incorporating the *Gen-CLIP* flow for training on generated images and the *CLIP* flow for inference on real images. This algorithm also details the cross-modality alignment loss and how we ensure alignment with large language models (LLMs) such as CLIPCap (Mokady et al., 2021), LLaVA (Liu et al., 2023), and LLaMA-3 (Meta, 2024).

Algorithm 1 summarizes the training and inference process for the GenRA framework, detailing how the model is trained on generated images using the *Gen-CLIP* flow, and subsequently applied to real images during inference. The algorithm also explains how to integrate aligned generated and real data with vision-language models such as CLIPCap, LLaVA, and LLaMA-3 for downstream tasks.

## C  MORE DISCUSSIONS ON GEN-REAL ALIGNMENT

In this section, we provide a comprehensive discussion of Gen-Real Alignment. Given training samples having the same text: real image $R$, synthetic $S$, and text $T$, let us denote our dual encoders as $f, g, h$ for real-encoder, syn-encoder, and text-encoder, respectively.

**Single vs. Dual Modality.** In a single-modality scenario (*i.e.*, a single encoder setup where $f = g$), given training would reduce distance $D(f(R), h(T))$ and $D(f(S), h(T))$, and then $D(f(R), f(S))$ would be reduced together. However, due to the nature of synthetic images, there could exist a gap between $R$ and $S$, such as unnatural artifacts, assuming $S$ contains spurious features. Therefore, under such approaches to put real and generated images into the same embedding space, generated artifacts may dominate, causing poor generalization and overfitting to synthetic patterns. Moreover, if the encoder ignores such different inputs $R$ and $S$, and produces representations that remain constant and equal, it can lead to "mode collapse" (LeCun, 2022; Assran et al., 2023), where the model overfits generated patterns, degrading performance on real data. On this line, we consider a dual-modality scenario, (*i.e.*, dual encoder setup where $f \neq g$) to prevent such a problem caused by reducing a distance $D(f(R), f(S))$. Here, we instead minimize $D(f(R), h(T))$ and $D(g(S), h(T))$, so allowing a small $D(f(R), h(S))$, not $D(f(R), f(S))$. Specifically, the expected role of $h$ is to ignore a synthetic complement of $S$ and produce representations that remain an intersection of $S$ and $R$ (having the same $T$). Such separate mappings of $f$ and $g$ would allow learning focused on shared characteristics between the real and generated modalities. Thereby treating generated images as a distinct modality, GenRA could prevent "mode collapse", enabling the effective use of synthetic data to augment real datasets without poor generalization and overfitting to synthetic patterns.

**Cross-Modality Alignment Loss.** Furthermore, the proposed cross-modality alignment loss aims to directly reduce a distance $D(f(R), h(S))$ allowing effective and faster training to convergence. As shown in Table 8, the proposed loss reduced training time and steps to convergence. Throughout our extensive experiments, for a given $R$ and $S$ having the same $T$, we have demonstrated the effect of minimizing a distance $D(f(R), h(S))$ which learns shared semantics between real and generated images while ignoring generated artifacts of $S$ may raise poor generalization on real images.

**Empirical Validation of Alignment Loss.** Nevertheless, we further conducted an ablation study on the effect of the cross-modality alignment loss (*i.e.*, the effects of directly reducing $D(f(R), h(S))$) under the dual encoder setup on COCO captioning. The results in Table 6 confirm that the alignment loss significantly bridges the modality gap, resulting in consistent performance improvements.

## D  MORE EXPERIMENTAL ANALYSIS

**Computational Costs.** We performed additional experiments to compare the computational costs. Table 8 are the updated results, including explicit details on the contributions of the cross-modality

Table 8: **Computational costs comparisons on COCO training.** Our GenRA introduces a slight increase in memory usage but remains more efficient on the convergence training time and steps than the baseline of indiscriminate mixing (gen+real) without alignment.

| Dual Projection | Alignment | Synthetic Data | Training Time (hrs) | Training Steps | Memory Usage (GB) | FLOPs (G) |
|:---:|:---:|:---:|:---:|:---:|:---:|:---:|
| ✗ | ✗ | ✗ | 8 | 50k | 24 | 70.2 |
| ✗ | ✗ | ✓ | 12 | 70k | 26 | 85.5 |
| ✓ | ✓ | ✓ | 10 | 60k | 28 | 85.5 |

Table 9: **Comparison with SigLIP on COCO captioning.** Our GenRA significantly improves SigLIP by effectively addressing the synthetic-real discrepancy. The best results are **bold**.

| Method | B@4 (↑) | CIDEr (↑) |
|:---|:---:|:---:|
| SigLIP | 37.51 | 117.82 |
| SigLIP + GenRA (ours) | **42.35** | **125.68** |

Table 10: **Visual question answering on ScienceQA.** We report the average accuracy on questions with the image context (IMG). The best results are **bold**.

| Method | Accuracy (%, ↑) |
|:---|:---:|
| LLaVA | 85.2 |
| LLaVA + GenRA (ours) | **87.6** |
| LLaMA-3 | 88.5 |
| LLaMA-3 + GenRA (ours) | **91.2** |

Table 11: **Comparison with models trained on real images.** We perform experiments on image captioning from pre-trained CLIP on generated images. The best results are indicated in **bold**.

| Dual Projection | Alignment | Fine-tuning Data | B@4 (↑) | CIDEr (↑) | SPICE (↑) |
|:---:|:---:|:---:|:---:|:---:|:---:|
| ✗ | ✗ | ✗ | 32.15 | 108.35 | 20.12 |
| ✗ | ✗ | Synthetic | 36.15 | 115.35 | 22.95 |
| ✓ | ✓ | Synthetic | **38.12** | **119.53** | **23.75** |
| ✗ | ✗ | Real | 38.24 | 119.78 | 23.86 |
| ✓ | ✓ | Real | **38.37** | **119.95** | **23.98** |

alignment loss and dual-model setup. The additional costs for GenRA stem from the cross-modality alignment loss, which facilitates aligning the features of generated and real images in a shared latent space, and the dual-projection setup, which processes the two modalities separately. Compared to CLIP without the dual projection, our GenRA introduces a slight increase in memory usage but remains more efficient on the convergence training time and steps than the baseline of indiscriminate mixing on generative and real data without the alignment.

**Comparison with SigLIP.** To strengthen the novelty of our work, we compared GenRA with SigLIP (Zhai et al., 2023) on COCO captioning. The results are shown in Table 9. SigLIP (Zhai et al., 2023) adopts a sigmoid loss for better image-text pre-training, focusing solely on real images. In contrast, our GenRA aligns real and generated images as distinct modalities, addressing the challenges of integrating synthetic data into training. GenRA is particularly relevant in scenarios requiring synthetic data, such as handling expensive attribute annotations or generating diverse samples. These results demonstrate that our GenRA complements SigLIP by effectively addressing the synthetic-real discrepancy, allowing for enhanced generalization and performance improvements.

**ScienceQA Results.** We also evaluated GenRA's performance on ScienceQA (Lu et al., 2022) when integrated with LLaVA (Liu et al., 2023) and LLaMA-3 (Meta, 2024). We calculated the average

Table 12: **Ablation study on LoRA rank and full fine-tuning.** We perform experiments on image captioning from pre-trained CLIP on generated images. The best results are indicated in **bold**.

| Method | B@4 (↑) | CIDEr (↑) | SPICE (↑) |
|---|---|---|---|
| LoRA (rank=2) | 36.85 | 117.62 | 23.10 |
| LoRA (rank=4) | **38.12** | **119.53** | **23.75** |
| LoRA (rank=6) | 37.96 | 119.12 | 23.60 |
| Full fine-tuning | 37.50 | 118.95 | 23.50 |

Table 13: **Quantitative similarity metrics comparisons on COCO.** We computed the cosine similarity between paired real and generated embeddings without and with alignment.

| Alignment | Cosine Similarity (↑) |
|---|---|
| ✗ | 0.52 |
| ✓ | 0.89 |

accuracy of questions with the image context. The comparison results are reported in Table 10. These results highlight GenRA's ability to improve generalization across multimodal tasks.

**Training on Real Images.** To illustrate the impact of GenRA on mitigating over-reliance on synthetic data, we compared performance on COCO captioning using real-only, mixed real-generated data, and GenRA alignment strategies. The results are shown in Table 11. These results indicate that GenRA's alignment strategy not only bridges the synthetic-real gap but also improves models trained exclusively on real data.

**Ablation on LoRA.** LoRA allows efficient adaptation to synthetic data while preserving the knowledge from pre-training on large-scale real data. This avoids the need for full fine-tuning, which can overwrite important pre-trained weights, especially when synthetic data is noisy or biased. The ablation results are reported in Table 12. As can be seen, LoRA with rank 4 achieves the best performance, balancing computational efficiency and alignment quality. Meanwhile, LoRA updates 35% fewer parameters compared to full fine-tuning while achieving better performance.

**Quantitative Similarity Metrics.** We quantified alignment using cosine similarity between paired real and generated embeddings on COCO dataset. The results are shown in Table 13. These results demonstrate that the alignment loss effectively bridges the gen-real gap, ensuring better feature consistency across modalities.

**Qualitative Embeddings Visualization.** To further validate the alignment between real and generated data, we conducted t-SNE (van der Maaten & Hinton, 2008) visualizations and cosine similarity analyses of the embeddings without and with alignment. Figure 3 shows the t-SNE plots of real and generated embeddings from 1000 samples in the COCO dataset. Without alignment, real and synthetic embeddings form two distinct clusters, reflecting the modality gap. With alignment proposed in our GenRA, the gap between real and synthetic embeddings is significantly reduced, with both modalities aligning closely.

# E QUALITATIVE VISUALIZATIONS

In this section, we provide qualitative visualizations of the generated images used in our experiments. Figures 4, 5, 6, 7, 8 and 9 show examples of images generated by Stable Diffusion (Rombach et al., 2022), alongside their corresponding real-world counterparts from the COCO dataset (Lin et al., 2014). Our visualizations demonstrate that the generated images closely resemble real images, capturing key semantic details and structural elements. However, subtle differences in texture or object placement are occasionally present. These artifacts highlight the importance of our Gen-Real Alignment (GenRA) framework, which ensures that these differences do not lead to model collapse by aligning the feature representations of generated and real images in the latent space. These visualizations further validate the effectiveness of our alignment strategy, ensuring that both generated and real data contribute equally to the model's understanding during inference.

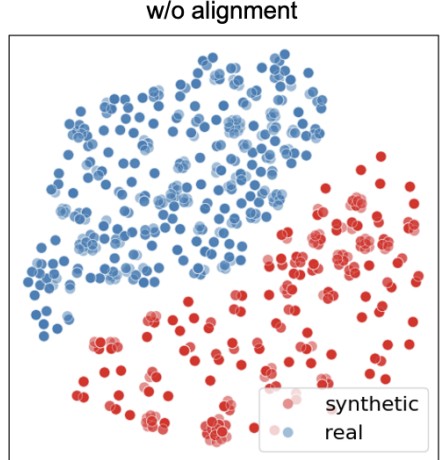 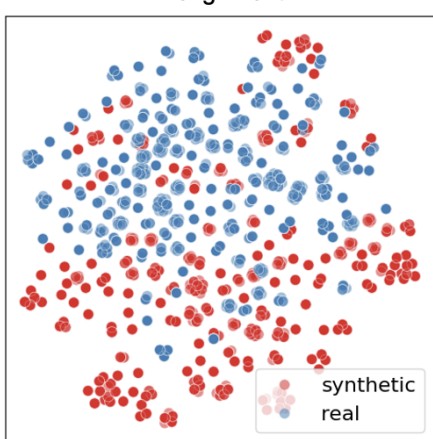

Figure 3: **Qualitative Visualizations of embeddings of real and synthetic images without (Left) and with (Right) alignment.** Blue and red dots denote the embeddings for real and synthetic images, respectively. Our GenRA with alignment significantly reduced the gap between real and synthetic images, with both modalities aligning closely in the latent space.

## F DISCUSSIONS

### F.1 LIMITATIONS

While our proposed GenRA framework shows significant improvements in aligning generated and real images, there are limitations to be addressed. The quality of the generated images is highly dependent on the underlying generative models, such as Stable Diffusion (Rombach et al., 2022). In scenarios where the generative model fails to produce realistic images, the alignment process may be less effective, leading to suboptimal performance in downstream tasks. Additionally, our method introduces additional computational overhead during the fine-tuning process due to the need for separate training on generated and real images, which may be a challenge in resource-constrained environments.

### F.2 BROADER IMPACT

Our work presents a novel approach to utilizing generated images for training vision-language models, offering a cost-effective and scalable solution for improving model performance. The use of generated data can reduce the reliance on real-world datasets, which are often expensive and time-consuming to collect. This has the potential to democratize access to high-quality training data for researchers and practitioners with limited resources. However, it is important to acknowledge the ethical concerns around the biases that can be introduced through synthetic data, especially if the generative models themselves are trained on biased datasets. We encourage future work to explore methods for mitigating these biases to ensure that the benefits of synthetic data can be realized in a responsible and equitable manner.

### F.3 MORE DISCUSSIONS

**Relevance Between Gen-Real Discrepancy and Model Collapse.** GenRA focuses on addressing the misalignment between synthetic and real data distributions during training. By aligning generated and real data in a shared latent space, GenRA enables the safe and effective integration of synthetic data for model training. Model collapse in (Shumailov et al., 2024) refers to the degradation of performance caused by recursive training on synthetic data (*e.g.*, models generating data that are then used for further training). This leads to a compounding drift from the real data distribution. To prevent model collapse in recursive scenarios, it is essential to first solve the gen-real discrepancy problem. Without addressing this gap, recursive training on synthetic data exacerbates the divergence between synthetic and real data distributions, accelerating model collapse. GenRA lays

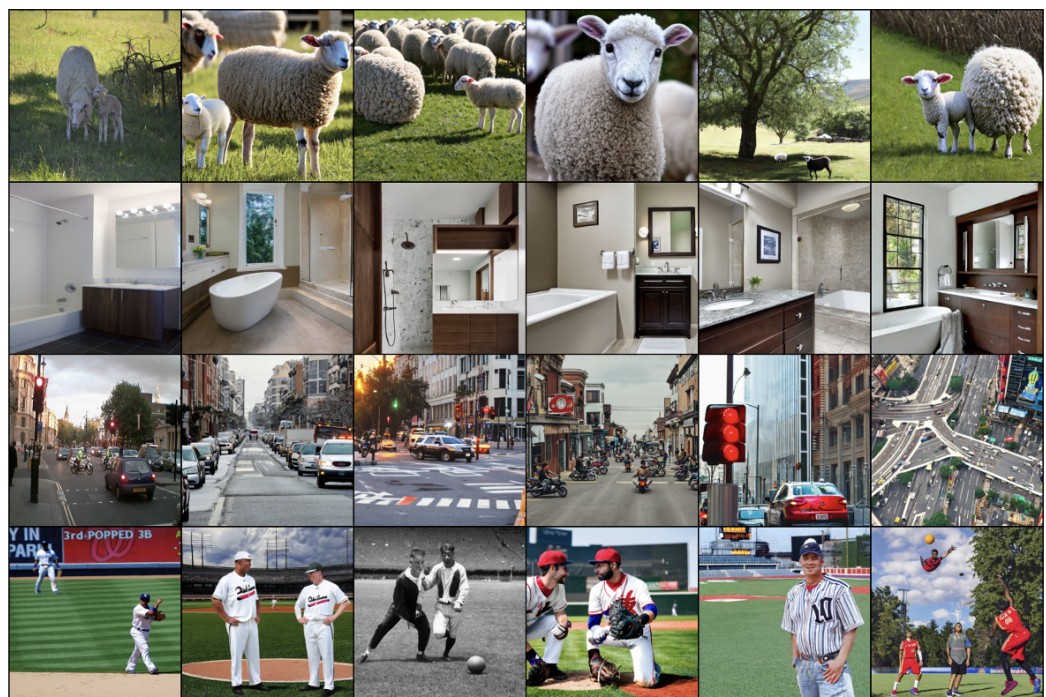

Figure 4: **Visualizations of real (Column 1) and generated images (Columns 2-6) using the same caption.** Those generated images generally capture high-level semantics in real images.

the groundwork by providing a robust framework for safely using synthetic data in non-recursive training scenarios.

**Applicability of GenRA to Recursive Training Scenarios.** While GenRA was designed for single-stage training using synthetic data, its principles could extend to recursive training: In recursive scenarios, each generation step could incorporate GenRA to realign synthetic data with real data. This would mitigate the compounding divergence that leads to collapse. By maintaining alignment at each stage, GenRA can act as a regularizer, ensuring synthetic data does not drift too far from real distributions over recursive iterations.

**Path Toward Escaping Model Collapse.** Our GenRA clearly articulates this pathway and plays the foundational role in safely integrating synthetic data, providing a step toward solving the broader model collapse problem.

- Step 1 (Our Work): Address gen-real discrepancies to ensure synthetic data can be safely used in training alongside real data.
- Step 2 (Future Work): Extend alignment techniques like GenRA to recursive training settings, where models rely entirely on synthetic data for iterative training and generation.

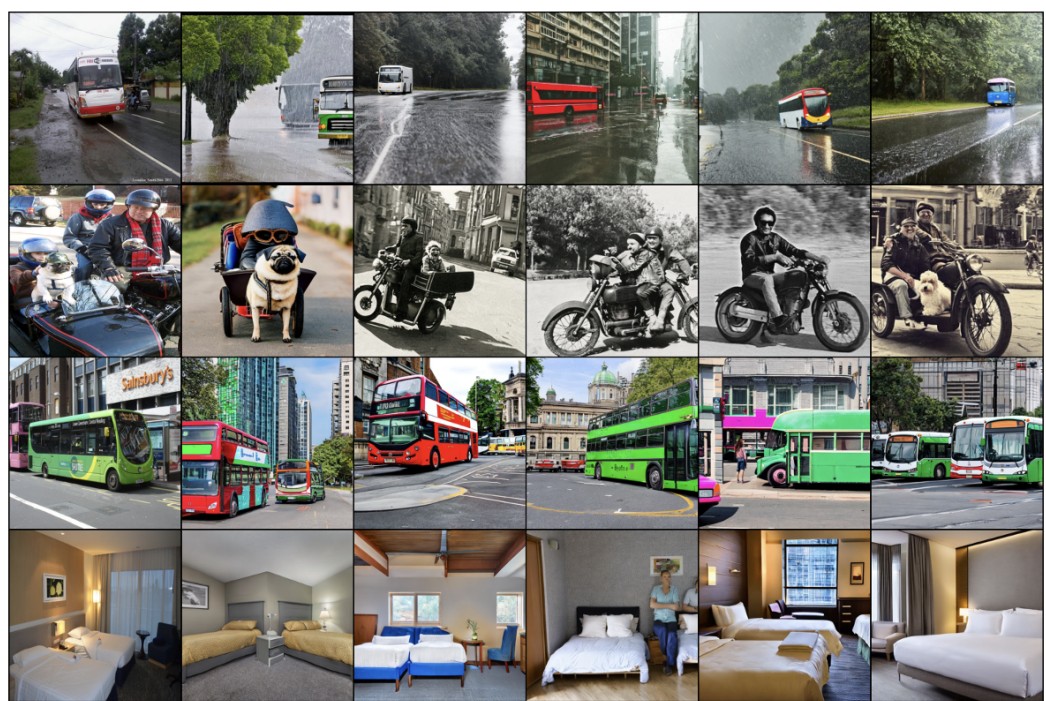

Figure 5: **Visualizations of real (Column 1) and generated images (Columns 2-6) using the same caption.** Those generated images generally capture high-level semantics in real images.

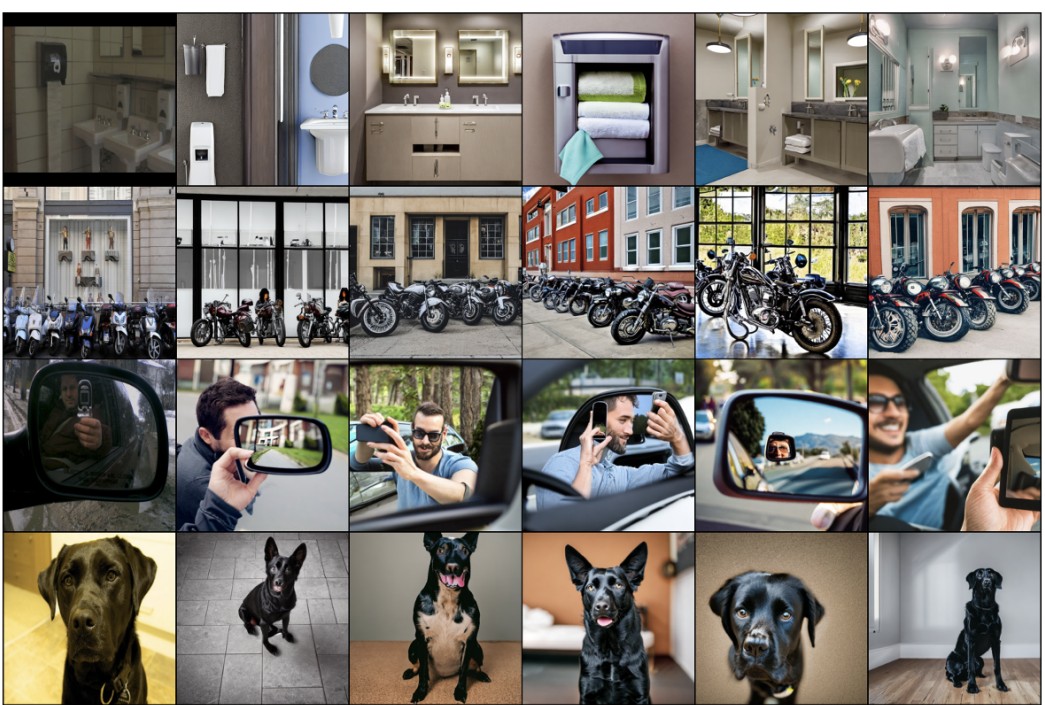

Figure 6: **Visualizations of real (Column 1) and generated images (Columns 2-6) using the same caption.** Those generated images generally capture high-level semantics in real images.

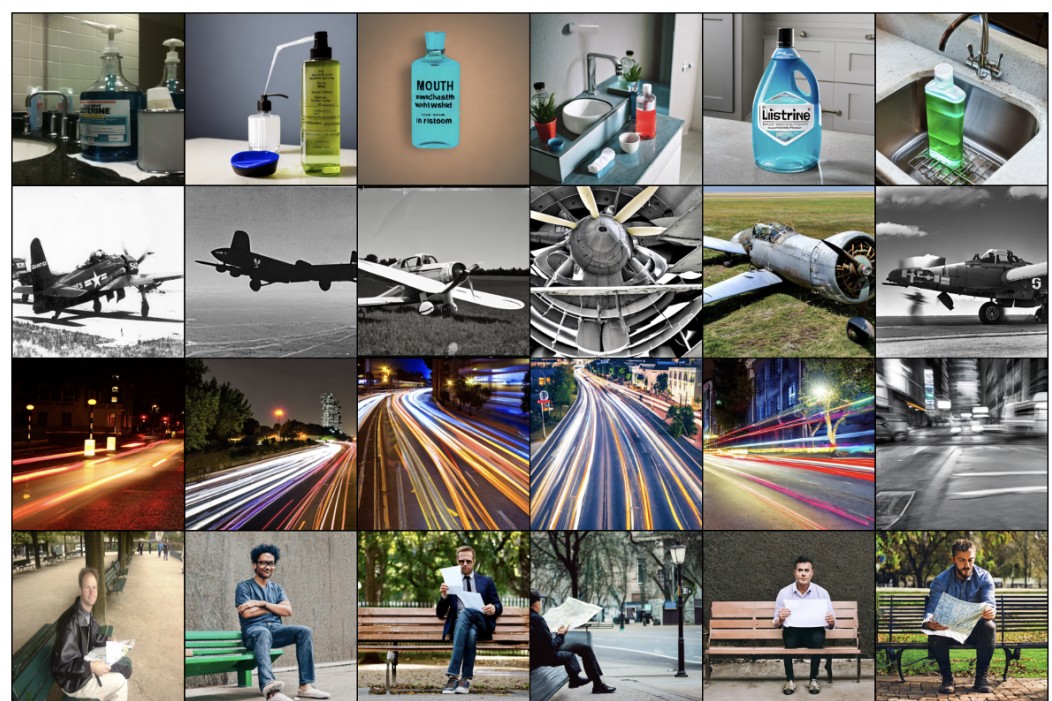

Figure 7: **Visualizations of real (Column 1) and generated images (Columns 2-6) using the same caption.** Those generated images generally capture high-level semantics in real images.

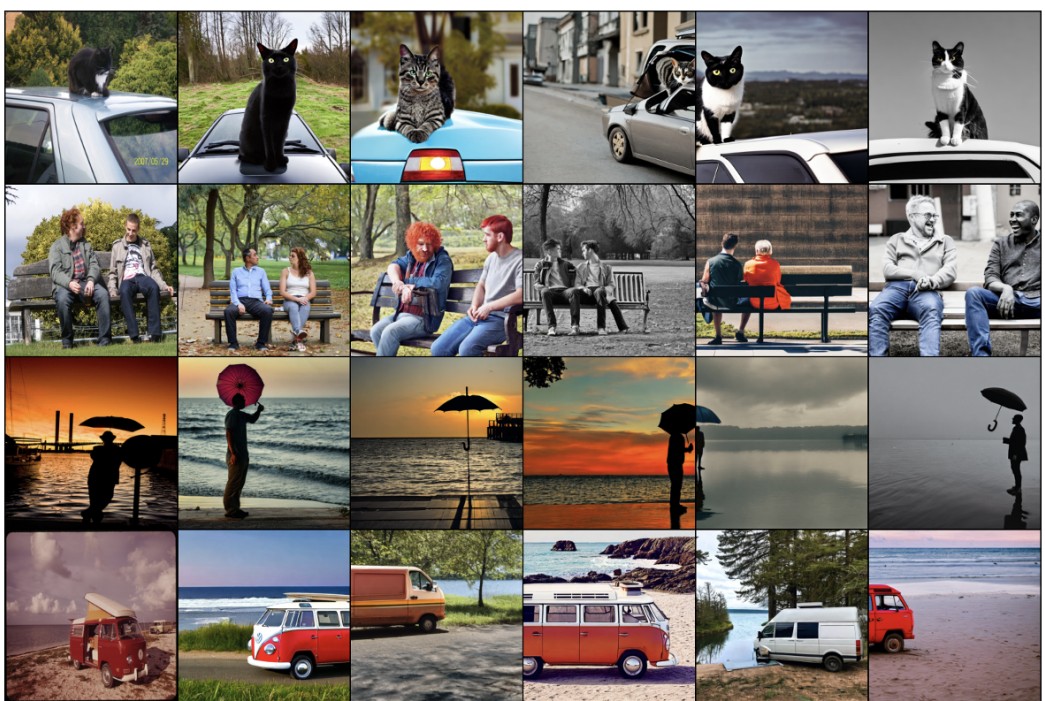

Figure 8: **Visualizations of real (Column 1) and generated images (Columns 2-6) using the same caption.** Those generated images generally capture high-level semantics in real images.

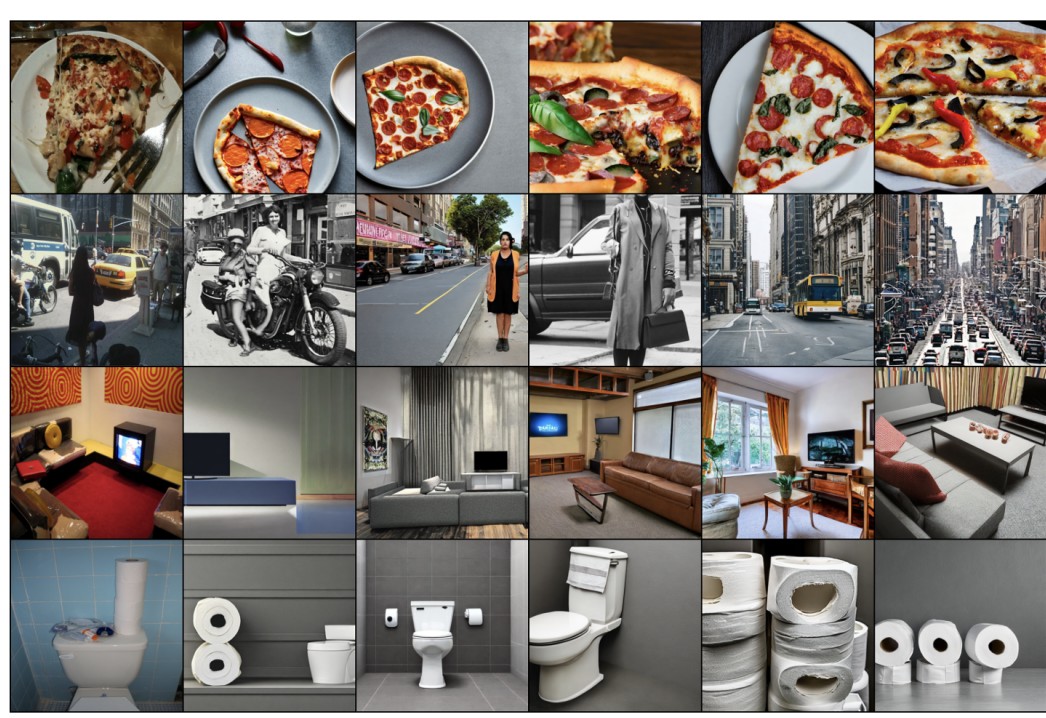

Figure 9: **Visualizations of real (Column 1) and generated images (Columns 2-6) using the same caption.** Those generated images generally capture high-level semantics in real images.

