# OpenReview forum: "Toward Escaping Model Collapse: Aligning Generated Images as a New Modality"
_ICLR.cc/2025/Conference — Submitted to ICLR 2025_

### Official Review · Reviewer_d89z · 2024-11-02

**Soundness:** 2
**Presentation:** 2
**Contribution:** 2
**Rating:** 6
**Confidence:** 4

**Summary:**

This paper aims to mitigate the distribution discrepancy between generated and real data, which leads to unsatisfactory performance when training solely on generated images. The authors propose a Generated-Real Alignment (GenRA) scheme that uses an extra low-rank adapter to aid the model in processing generated images, which is trained firstly by the contrastive loss between generated images and texts and the subsequent contrastive loss between generated and real images. Experimental results on multiple benchmarks show considerable improvements of GenRA over the baseline methods.

**Strengths:**

1. The paper is well-motivated and studies an important problem. Training with synthetic data has attracted mounting attention over the past year. Despite the significant advancement of generative models, they still cannot exactly model real data distribution. Addressing this problem is important for alleviating human labor in data curation.

2. This work proposes to use an extra LoRA to accommodate the syn-to-real discrepancy, which is a novel way to address this problem to the best of my knowledge.

3. Evaluations on multiple benchmarks and multiple tasks (e.g., retrieval, classification, and captioning) exhibit promising results for the proposed GenRa.

**Weaknesses:**

1. The presentation of the method is not clear enough for the audience. Specifically, in Figure 1, there are two vision encoders/projectors for the model, one for the real images and the other for the synthetic ones. Are they essentially the same model? If so, why do the authors emphasize 'dual-encoder' in L214? It is therefore not clear to me which encoder the proposed method uses during inference for the real image.

2. The title somewhat over-claims the actual contribution of this work. The proposed method targets mitigating the syn-real discrepancy but not the mode collapse problem when training solely or recursively on the synthetic data. GenRA trains with real images (as in Sec. 3.3) and does not involve recursive data generation and consumption, which is not the same case as the mode collapse problem in the referred paper [Shumailovetal.,2024].

3. The role of LoRA is not well ablated. LoRA plays a crucial role in the GenRA, how would the rank of LoRA affect the performance? how would LoRA compare with the full fintuning?

**Questions:**

It would be better to have qualitative and quantitative studies on the alignment of the features of the real and synthetic images, e.g., TSNE visualization and similarity metrics.

---

### Official Review · Reviewer_MMVr · 2024-11-03

**Soundness:** 3
**Presentation:** 3
**Contribution:** 4
**Rating:** 6
**Confidence:** 4

**Summary:**

This paper introduces a novel framework aimed at addressing modality discrepancies between real and generated images when used in training machine learning models. The authors propose treating generated images as a separate modality rather than substituting them directly for real images, which often leads to "model collapse" or performance degradation. The framework, termed Generated-Real Alignment, applies a cross-modality alignment loss to fine-tune a model specifically on generated images, creating a shared latent space where both real and generated images can be effectively integrated. This approach appears versatile, being compatible with a range of vision-language models, and achieves improved performance on multiple tasks, such as image captioning, zero-shot image retrieval, and image classification.

**Strengths:**

Clear Problem Identification: The authors recognize and address a significant issue in the use of generated images as training data, which has become more relevant with the advancements in generative models. By framing the problem as a "modality discrepancy," the paper effectively motivates the need for a novel approach.
Innovative Approach: Introducing generated images as a distinct modality and aligning them with real images through a dedicated loss function is a unique concept that has potential practical implications. The authors' method appears to be an improvement over indiscriminate data blending.
Comprehensive Experiments: The paper’s experimental design covers a range of tasks, showing generalizable improvements across different vision-language models and tasks. This strengthens the case for the approach's robustness and adaptability.
Scalability: The framework’s demonstrated compatibility with various models and tasks suggests it may have broad applications in the field of vision-language learning.

**Weaknesses:**

Limited Theoretical Analysis: While the experimental results are promising, there is limited discussion on why the cross-modality alignment loss is particularly effective in bridging the modality gap. Adding a more detailed theoretical justification for the approach could make the contribution more substantial.

**Questions:**

Clarity on "Model Collapse": The paper could clarify what constitutes "model collapse" in this context, perhaps with concrete examples or metrics. This would help readers understand the severity of the issue the proposed method aims to mitigate.

---

### Official Review · Reviewer_4Pkz · 2024-11-04

**Soundness:** 3
**Presentation:** 3
**Contribution:** 2
**Rating:** 5
**Confidence:** 4

**Summary:**

The paper introduces GenRA (Generated-Real Alignment), a framework designed to address the challenges of integrating generated images into machine learning training pipelines. Generated images, although realistic, often differ from real images in subtle ways, potentially leading to model collapse. GenRA treats generated images as a separate modality and aligns them with real images in a shared latent space to address this. The framework fine-tunes a pre-trained model on generated images using a cross-modality alignment loss, improving performance across various vision-language tasks. Extensive experiments demonstrate the effectiveness of this approach on tasks such as image captioning, zero-shot retrieval, and classification.

**Strengths:**

1. GenRA introduces an approach by treating generated and real images as distinct modalities, bridging the gap through explicit alignment in a shared latent space. This is a unique way of tackling the modality discrepancy issue.
2. The authors provide extensive experimental results across multiple benchmarks, including image captioning and zero-shot retrieval, demonstrating significant performance improvements.
3. The framework’s effectiveness improves as the scale of generated data increases, making it suitable for large datasets like CC12M and highlighting its potential for broader applications.

**Weaknesses:**

1. The addition of the cross-modality alignment loss and the dual-model setup introduces computational complexity, possibly limiting efficiency. Could the authors compare the training cost of the proposed method? For example, the authors can list the training time, training steps, memory usage, or FLOPs against baseline methods.

2. This method's training details are not quite clear, especially regarding the construction of the training data and the training step settings for each process. Please show more details, including the number of generated images used for training, training batch size, the number of steps for training "proj for real" and "proj for gen".

3. Although the experimental results have improved significantly, I still doubt whether the novelty is enough since the main contribution is to improve the feature extraction.
(1) Maybe comparing with SigLIP [1r] could strengthen the novelty. Please explain how the proposed method differs conceptually from SigLIP, or discuss why the comparison would be relevant given the different focus of the two approaches (generated image alignment vs. general image-text pretraining).
(2) The extraordinary abilities of LLaVA and LLaMA-3 could include question answering and visual question answering. It would be better to include their performance on these tasks.
(3) Please clarify how the contribution goes beyond just improving feature extraction.

4. Common sense suggests that real images are often better than synthetic images. So, I am wondering whether the proposed method would achieve better performance if using real images. It would be interesting to compare it with another baseline that is trained with real images.

5. Minor:
 (1) Citation error in Line 351: LLAMA-3 (?)

[1r] Zhai, Xiaohua, et al. "Sigmoid loss for language image pre-training." Proceedings of the IEEE/CVF International Conference on Computer Vision. 2023.

**Questions:**

Please address my concerns above. Thank you.

---

### Meta-Review · Area_Chair_aJLL · 2024-12-19

**Metareview:**

This paper proposes a framework, named GenRA (Generated-Real Alignment), to mitigate the modality discrepancies between real and synthetic images through a multi-modal learning approach. Extensive experiments across multiple benchmarks are provided, and scaling capability is also shown on larger datasets. However, reviewers pointed out the lack of novelty of the method, and a detailed discussion on applying the method to real datasets is still missing. Overall, it seems the contribution and novelty of this work are limited. Therefore, based on the reviews, I would not recommend accepting the paper.

**Additional Comments On Reviewer Discussion:**

Reviewer 4Pkz asked for the details of training, comparison with SigLIP, and results on real images. During the rebuttal, the authors provided extra evidence on the above questions. It seems some detailed analysis is still lacking.

Reviewer MMVr asked for clarification on model collapse and more theoretical analysis, and the authors did a good job of making a sound and solid explanation with experimental results.

Reviewer d89z asked for better writing of the method and ablation of the loRA, which were provided by the authors in the rebuttal.

---

### Decision · Program_Chairs · 2025-01-22

Reject